# Visualization Method for Arbitrary Cutting of Finite Element Data Based on Radial-Basis Functions

**Shifa Xia [1,2], Xiulin Li [1,2]** 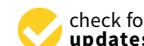 **, Fuye Xu [3], Lufei Chen [3] and Yong Zhang [3,\***

[1]   Division of Materials, China Institute of Water Resources and Hydropower Research (IWHR), Beijing 100038, China
[2]   State Key Laboratory of Simulation and Regulation of Water Cycle in River Basin, China Institute of Water Resources and Hydropower Research (IWHR), Beijing 100038, China
[3]   Beijing Key Laboratory of Multimedia and Intelligent Software Technology, College of Metropolitan Transportation, Beijing University of Technology, Beijing 100124, China
\*   Correspondence: zhangyong2010@bjut.edu.cn; Tel.: +86-10-67396568-2103

**Abstract:** Finite element data form an important basis for engineers to undertake analysis and research. In most cases, it is difficult to generate the internal sections of finite element data and professional operations are required. To display the internal data of entities, a method for generating the arbitrary sections of finite element data based on radial basis function (RBF) interpolation is proposed in this paper. The RBF interpolation function is used to realize arbitrary surface cutting of the entity, and the section can be generated by the triangulation of discrete tangent points. Experimental studies have proved that the method is very convenient for allowing users to obtain visualization results for an arbitrary section through simple and intuitive interactions.

**Keywords:** radial basis function (RBF) interpolation function; finite element data; arbitrary cutting; visualization

---

## 1. Introduction

In the field of engineering technology, there are many common scientific calculation methods, among which the finite element method is the most widely applied. It is a general approximate calculation method for solving mathematical equations. On the other hand, scientific computing visualization is a new technology that converts digital information into image information. With rapid development of the society and the large increasing amounts of data generated every day, a growing number of scholars are selecting the finite element analysis method as their theoretical basis and they create a more convenient tool to simplify the analysis and collation of finite element data by integrating visualization technology.

Finite element data visualization methods are primarily divided into two types: surface rendering and volume rendering [1]. The surface rendering method can only render the outer surface and cannot meet the requirement for displaying the data inside an entity. The volume rendering method can show the overall state of data; however, with the large-scale growth of data, the images are occluded from each other, therefore it is not easy to show unclear internal problems. Generally, there are two methods to solve this type of problem. The first defines multiple transformation functions for a data field by reducing the transparency of a part of the data; in this manner, the internal conditions can be shown [2]. The second is based on the cutting method [3], which can display the occluded area in a more intuitive manner. Cutting method is divided into plane cutting and arbitrary closed polyhedron cutting. The plane cutting algorithm is a simple one that can only display a flat surface after cutting. The polyhedron cutting algorithm is more complex with professional operations usually in need.

---



To analyze the internal data of the finite element entity, the finite element data are decided to be split. Radial basis function (RBF) interpolation is a combination of a series of precise interpolation methods, that is, the interpolation surface must pass through each measured sample value. RBF is a function that varies with the distance from a certain location. As an accurate interpolator, the RBF method is different from global and local polynomial interpolators. They all are not precise interpolators (they do not require the surface to pass through the measuring point). Compared with RBF, Inverse Distance Weighted (also precise interpolator) never predicts values greater than or less than the maximum measurement value, while RBF predicts values greater than or less than the minimum measurement value, and can generate smoother surfaces through discrete points. In Turk et al.'s study, RBF interpolation is applied to generate arbitrary internal sections for volume data. After cutting, their method not only shows the entity but also visualization results such as the surface of any section and contour lines. Inspired by Turk et al.'s work, we tried to obtain arbitrary internal sections via RBF interpolation.

## 2. Related Work

Finite element data visualization comprises surface rendering and volume rendering. The basic concept of surface rendering method is approximating the object surface by using planar meshes. The specific method involves pre-processing the data to obtain a triangular surface, then using polygon approximation for fitting, and finally using a graphics algorithm to draw the relevant surface. There are two main methods for surface rendering. The first is a polygon-oriented one [4], a method for which the surface contour is calculated and then the object surface is spliced according to the contour. The second method is grid-oriented [5–7], which uses triangle splicing to form surfaces, but these triangles are located in a single cube element. According to the adjacent topological relationship of the cube element surface, a polygon data set is constructed and the surface is then generated quickly. However, when the shared surface of neighboring volume element has four intersections, surface calculation errors will occur. In response to issues such as ambiguity surfaces, Cline et al. decomposed the cube element to the size of a pixel, directly drew the surface with pixel points, and proposed the decomposition cube method [8]. The more common surface rendering applications includes contour lines, color cloud maps, and slices. Although the surface rendering method appears earlier and is widely used, it can only display the surface without showing the internal information of the data field.

To display the three-dimensional data inside and outside the objects from multiple directions, the volume rendering method gradually became more popular [9–12]. It is a grid-based method that can display the surface and internal volume metadata separately as well as mixed data. The volume rendering method starts from each grid, according to the different properties of each grid, by using the most suitable illumination model. After traversal, each grid is assigned to different color and distinct transparency; then, according to the principle of optical energy concentration when light penetrates a translucent material, it will ultimately perform color fusion. Compared with surface rendering, volume rendering achieves a more comprehensive rendering effect and can display subtle and hard-to-define features in the data field without generating intermediate geometric primitives. However, with the increasing amount of data, volume rendering methods will consume more resources, and it will be difficult to complete such large-scale rendering on an ordinary PC. Even for professional graphics service equipment, the required rendering time is extensive.

To display the data inside the entity, some researchers have proposed some novel methods like reducing the transparency of a part of the data set, plane cutting and polyhedron cutting. Wang et al. used the backward projection method to calculate the coordinates of each point of the section surface in any direction and obtained a display plane with a clear cut in any random direction [13]. Qi et al. optimized the plane cutting algorithm from the perspective of interaction [14], and Westermann et al. used three-dimensional texture hardware to accelerate the implementation of polyhedron-to-volume data cutting [15]. Weiskopf et al. implemented the depth-based cutting technique to determine the regions that need to be cut according to the depth structure of the polyhedron boundary [16].

Erickson et al. realized the acceleration of the cutting algorithm by minimizing the number and length of polyhedron cutting edges, therefore greatly reducing the time complexity of the algorithm [17]. Although there are many existing cutting algorithms, they are slightly inadequate in different aspects. The common disadvantages of plane cutting and polyhedron cutting are tedious interaction and poor user experience.

## 3. Overview of the Visualization Method for Arbitrary Cutting of Finite Element Data Based on RBF

In our method, to reduce the waiting time of the system, a surface-based rendering method is adopted to meet the general requirement of rendering only the outer surface of the entity. In addition, to display the internal data of finite elements, this study implements an arbitrary cutting method for finite element data. With the rapid development of rendering technology and hardware equipment, as described by Qi et al. [14], interactive experience has become the main bottleneck for large-scale three-dimensional visualization tools. Zhang et al. proposed an improved parametric line clipping and polygon clipping algorithm to complete the cutting [18], while we used surface clipping. Takayama et al. cuts the model by drawing a freeform stroke [19]; they mainly focused on the generation of texture. Inspired by their interaction mode, we intend to use the interactive method of drawing free curves to complete data cutting. To improve the user experience of the software and simplify the interactive process, this study applies RBF interpolation to finite element data cutting and proposes a visualization method for arbitrary curve section. In contrast to traditional methods, our method supports not only plane cutting, but also arbitrary surface cutting; in our method, the user can achieve an arbitrary screen curve via drawing. After that, the smooth function is constructed using the RBF interpolation method, followed by the calculation of the value of finite element data in the function. Then, coordinate transformation will be executed, due to different coordinate systems. After performing the cutting algorithm, discrete cutting points are calculated. Finally, the cutting points are triangulated to draw the entities after cutting. Figure 1 shows the entire process of our method.

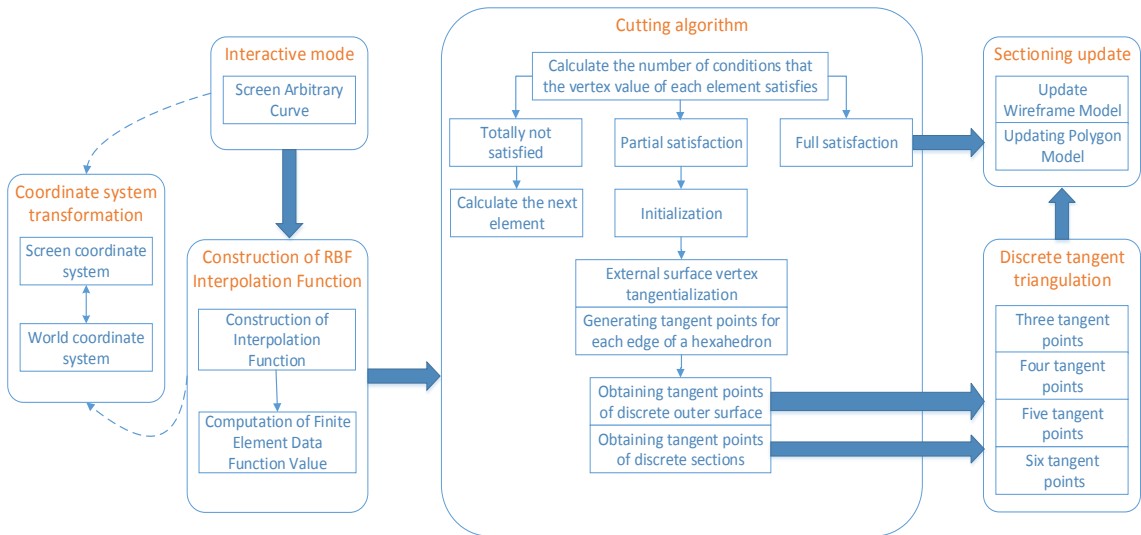

**Figure 1.** Visualization method for arbitrary cutting based on radial basis function (RBF).

## 4. Interaction Mode

The traditional cutting methods always use a plane to construct the section and they can only obtain a plane section. However, the plane section can not satisfy the analysis of internal data. Although there are some methods to obtain arbitrary cutting surfaces, most of them require professional operations. Therefore, the interaction is very tedious, and the cutting surface cannot be obtained quickly.

To solve these problems and further improve the interactive experience, we have proposed a very simple interaction method. Users can obtain arbitrary cutting surfaces easily by drawing arbitrary curves on the screen. This interactive method can specify arbitrary cutting surfaces, and users can cut solid models by drawing free curves. Because the discrete points of the free curve are located on the screen coordinate system, the data vertex region division can be realized by transforming the finite element data into the screen coordinate system. As shown in Figure 2, the finite element data are divided into three types by the free curve. The first is the points above the curve; the second is the points below the curve; the third is the points on the curve. In addition, to draw the free curve of the screen, the discrete points must be transformed into the world coordinate system through transformation, and the distance parameter must be adjusted to draw the free curve in front of the entity and make its viewpoints closer; in this manner, mutual occlusion between entities and curves can be avoided. Through the free curve, a set of two-dimensional coordinate points on the screen can be obtained, which provides the necessary input for the restrictions of the RBF smooth interpolation function which will be mentioned later.

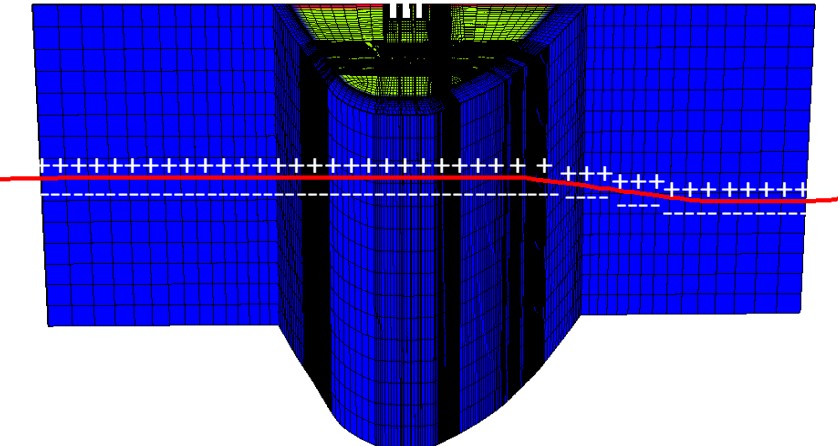

**Figure 2.** Arbitrary curve cutting mode.

## 5. RBF Interpolation Function Construction

To clarify the position relationship between the vertices of the finite element data and the free curve, by using the two-dimensional coordinate set on the screen, the constraint set and real number set are constructed, and the function equation is obtained by further interpolation fitting. The interpolation process can be described by the provided constraint set $C = \{c_1, c_2 \ldots, c_k\}$ containing K different points and the corresponding real number set $H = \{h_1, h_2 \ldots, h_k\}$; we have to find a smooth function F(x) that satisfies the equation

$$F(c_i) = h_i, 1 \leq i \leq k. \tag{1}$$

To solve this problem, we first need to obtain the constraint set $C$ at different points and the corresponding real number set $H$. After the user draws any curve on the screen, we will obtain a discrete set of screen coordinates $P = \{p_1, p_2 \ldots, p_k\}$, where one of the screen coordinates can be expressed as

$$p_i = (x_i, y_i), 1 \leq i \leq k. \tag{2}$$

Herein, the necessary input data of equation F(x) is obtained by calculating the set $P$ of screen discrete points. For each adjacent screen coordinate point, the two-dimensional vector is obtained in this study as follows:

$$\vec{v_i} = p_{i+1} - p_i = (x_{i+1} - x_i, y_{i+1} - y_i) = (x_i', y_i'). \tag{3}$$

We calculate its module as follows:

$$\| \overrightarrow{v_i} \|_2 = \sqrt{x_i'^2 + y_i'^2}. \tag{4}$$

Finally, after the normalization of the direction of the vector $E = \{e_1, e_2 \ldots, e_k\}$, Formulas (3) and (4) can be used to calculate $E$, and $\overrightarrow{e_i}$ is given by

$$\overrightarrow{e_i} = \frac{\overrightarrow{v_i}}{\| \overrightarrow{v_i} \|_2} = \frac{(x_i', y_i')}{\sqrt{x_i'^2 + y_i'^2}} = (\hat{x}_i, \hat{y}_i). \tag{5}$$

According to the normalized direction vector $E$, we find the coordinates $c_{2i-1}$ and $c_{2i}$ of two points with equal distances perpendicular to the vector and both pairs of coordinates corresponding to all direction vectors $E$ form the constraint set $C$ for the points. For the two-dimensional coordinate situation of the screen, $c_{2i-1}$ and $c_{2i}$ are given by Formulas (6) and (7), respectively:

$$c_{2i-1} = p_i + \alpha \overrightarrow{q_i}, \tag{6}$$

$$c_{2i} = p_i + \alpha \overrightarrow{\hat{q}_i}. \tag{7}$$

These lead to $\overrightarrow{q_i}$ and $\overrightarrow{\hat{q}_i}$, respectively, in Formula (8) and Formula (9):

$$\overrightarrow{q_i} = (-\hat{y}_i, \hat{x}i), \tag{8}$$

$$\overrightarrow{\hat{q}_i} = (\hat{y}_i, -\hat{x}i). \tag{9}$$

Formula (8) and Formula (9) respectively represent two two-dimensional vectors pointing in two different directions perpendicular to the direction vector $\overrightarrow{e_i}$, and $\alpha$ represents the distance perpendicular to the direction vector $\overrightarrow{e_i}$. The real number of $h_i$ at point $c_{2i-1}$ above $\overrightarrow{e_i}$ is 1, the real number of $h_{i+1}$ at point $c_{2i}$ below $\overrightarrow{e_i}$ is –1, and the solution for the constraint set and the real number set is shown in Figure 3. Through the above method, the constraint set $C$ at different points and the corresponding real number set $H$ are finally determined, providing the interpolation qualification condition for the function F(x).

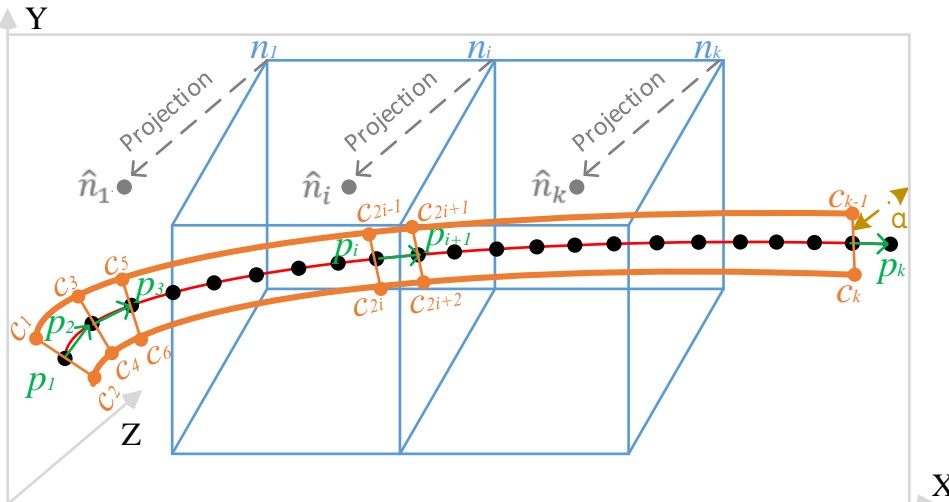

**Figure 3.** Schematic of RBF interpolation.

After the interpolation condition is obtained, the radial basis function can be used to generate the smooth equation. The radial basis function is a real-valued function whose value only depends on the distance of any point c. The plate spline- based function is a commonly used interpolation method to fit the bivariate smooth function. It searches for a smooth surface with minimum bending through all

control points, that is, to fit the constraint set *C* with the minimum curvature surface. The plate spline-based function is shown below:

$$\Phi(x) = |x|^2 log(|x|).\tag{10}$$

According to Formula (10), the smoothing function F(x) can be rewritten into equation:

$$F(x) = \sum_{j=1}^{n} d_j \phi(x - c_j) + P(x).\tag{11}$$

In the above equations, $c_j$ represents the position of the constraint set, and $d_j$ is the weight. P(x) is a polynomial of the linear constant F. By determining the weight $d_j$ and the coefficient of P(x), a radial basis function can naturally meet the given interpolation constraints. Therefore, this method provides an accurate solution that can ensure that the finite element analysis method is not affected excessively by the approximation and discretization errors. To obtain the weight set $d_j$ under the condition of satisfying the interpolation condition in Formula (1), it is able to obtain Formula (12) by using Formula (11):

$$h_i = \sum_{j=1}^{k} d_j \phi(c_i - c_j) + P(c_i).\tag{12}$$

Because the coefficients of the equation are linear with respect to $d_j$ and P(x), they can be calculated using the linear method.

For the two-dimensional interpolation equation, assuming that $c_i = (c_i^x, c_i^y)$, $\phi_{ij} = \phi(c_i - c_j)$, then the linear system can be written in the following form:

$$\begin{bmatrix} \Phi_{11} & \Phi_{12} & \cdots & \Phi_{1k} & 1 & c_1^x & c_1^y \\ \Phi_{21} & \Phi_{22} & \cdots & \Phi_{2k} & 1 & c_2^x & c_2^y \\ \vdots & \vdots & & \vdots & \vdots & \vdots & \vdots \\ \Phi_{k1} & \Phi_{k2} & \cdots & \Phi_{kk} & 1 & c_k^x & c_k^y \\ 1 & 1 & \cdots & 1 & 0 & 0 & 0 \\ c_1^x & c_2^x & \cdots & c_k^x & 0 & 0 & 0 \\ c_1^y & c_2^y & \cdots & c_k^y & 0 & 0 & 0 \end{bmatrix} \begin{bmatrix} d_1 \\ d_2 \\ \vdots \\ d_k \\ p_0 \\ p_1 \\ p_2 \end{bmatrix} = \begin{bmatrix} h_1 \\ h_2 \\ \vdots \\ h_k \\ 0 \\ 0 \\ 0 \end{bmatrix}.\tag{13}$$

The linear system in Formula (13) is positive definite semi-symmetric; thus, $d_j$ and $p_j$ both have a unique solution. Here, $c_i$ represents the coordinates of the point i in the constraint set. $1 \le i \le k$ represents the number of points in the range of 1 to k, $c_k^x$ and $c_k^y$ respectively represent the constraint set of numbers for the x- and y-component coordinates of k points, and $\Phi(x)$ represents the radial basis function (RBF), where $x = c_i - c_j$ represents the distance between two points $c_i$ and $c_j$, $\Phi_{ik}$ represents the function of Euclidean distance between the numbers for the i point and numbers for the k point, $h_i$ represents the value of the function at the coordinates point i, $d_k$ denotes the weight, and P(x) is a polynomial of linear constant *F*. Building smooth equations can be understood as the process of setting all the coordinates of point $c_i$, under known constraints, with every two points of the RBF $\Phi_{ik}$ and function value $h_i$, to identify the unknown variable weights $d_j$ and a polynomial P(x).

By determining the weighting $d_j$ and a polynomial P(x), we can identify a smoothing function F(x) that satisfies the interpolation condition in Formula (1). As shown in Figure 3, F(x) = 1 represents the orange curve above the red screen curve connected by constraint points with odd subscripts; F(x) = −1 represents the orange curve below the red screen curve connected by constraint points with even subscripts, and F(x) = 0 represents the red screen free curve. After transforming the finite element vertex coordinates to the screen coordinate system, we can accurately determine the position relationship between each vertex and this line by using the function F(x).

## 6. Coordinate System Transformation

After the smooth function F(x) is obtained, it can be used to cut three-dimensional entity data. The smooth function is constructed in the screen coordinate system and needs to be transformed. The transformation process is mainly divided into the following steps:

- Determine the model coordinates according to the modeling function.
- Transform into the world space through modeling transformation.
- Transform to corresponding view coordinates via ModelView Transformation.
- Through projection transformation, corresponding viewport coordinates will be calculated.
- Transform viewport coordinates to screen coordinates, and finally get the pixel coordinates.

After the above processes, the model coordinate is transformed into the screen coordinate, and the transformation matrix can be written as follows:

$$MVPW = ModelView * Project * Window. \tag{14}$$

In Formula (14), *ModelView* represents the model viewpoint matrix that transforms the object from the world coordinate system to the camera coordinate system; *Project* represents a projection matrix that projects a three-dimensional object onto a two-dimensional plane; *Window* represents the mapping of an object in a two-dimensional plane to a window to complete the task of drawing the window matrix for display. With the help of the *MVPW* matrix, we can easily convert the model coordinate system to the screen coordinate system, or vice versa. The formula for the coordinate transformation can be expressed as follows:

$$WindowPt = WorldPt * MVPW, \tag{15}$$

$$WorldPt = WindowPt * MVPW^{-1}. \tag{16}$$

In Formula (15) and Formula (16), *WindowPt* represents the screen coordinates, *WorldPt* represents the coordinates of the model in the world coordinates system, the coordinates of *MVPW* represents the Formula (14), and $MVPW^{-1}$ for *MVPW* matrix inverse matrix.

For each point on the finite element entity, the transformed value is substituted into the smooth function F(x) after coordinate transformation, and the obtained value can only be greater than 0, less than 0 or equal to 0, denoting the point that at the interpolation function, the point above the interpolation function, and the point below the interpolation function, respectively. With the position relation between the finite element data and the interpolation function, each mesh can be easily cut to generate the cutting point.

## 7. Cutting Algorithm

After the values of the finite element data in the smoothing function F(x) are given, the cut entity and the section are visualized. During cutting, the vertices that need to be redrawn and the new cut points on the surface of the cut finite element mesh must be recorded. According to these data, the polygon model and wireframe model after cutting are updated. Our cutting algorithm mainly uses two Map containers, which is a standard data structure designed by STL(C++ Standard Template Library). They are outer surface Map and the cutting surface Map. In the Map, Key denotes the number of the current surface in the current grid cell in the cutting process, and Value represents the set of the vertex of the corresponding surface in the affected grid cell and the newly generated vertex. Among them, the outer surface Map container maintains the points on the outer surface corresponding to the finite element mesh that needs to be redrawn after cutting, and the other cutting Map container maintains the cutting points on the newly generated section of the finite element mesh after cutting.

First, all finite element mesh elements are traversed, and eight vertex indexes are obtained, according to the vertex index, to obtain the value of that point in the RBF interpolation function and

determine what points need to be redrawn; if all eight meet the requirements, we identify the ones belonging to the outer surface of the cell, according to the points on the surface of the index number, and update them after cutting the new polygon model and a wireframe model. If the requirements are not met, we directly skip and continue to judge the next finite element mesh. Otherwise, cutting occurs.

For the newly generated pointcut, part of the pointcut information is encapsulated to facilitate subsequent work, such as triangulation. In this study, the data structure of the pointcut is defined according to the algorithm requirements: a tangent point first needs to record its own ID and coordinate values in the *x*-, *y*-, and *z*-directions; then, the surface, edge, and point of the grid element to which the point belongs must be recorded to distinguish the point of tangency and the original vertices; the number of the tangent point is marked −1. The data structure of the tangent point is shown in Figure 4, where the number in green represents the surface number, the number in orange represents the edge number, and the number in black represents the point number. For example, a hexahedron has six faces, where 0–5 correspondingly represent the bottom, top, front, back, left, and right sides. Edges and points are also marked with these numbers.

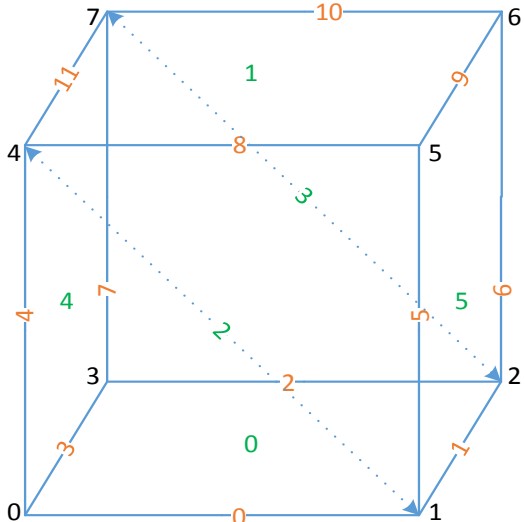

**Figure 4.** Data structures of tangent point.

We then perform the cutting algorithm. First, we initialize the outer surface Map and the cutting surface Map of the hexahedron and the coordinate value and key value of the eight points of the hexahedron. The key value here refers to the total number of vertices multiplied by the vertex index value plus the vertex index value to ensure uniqueness. Then, each face of the hexahedron is traversed to obtain the neighbors of that face; if there are no neighbors on the current face, then it is the outer surface. By traversing the four points of the outer surface, the values of these four points in the smooth function F(x) are obtained. If the conditions are met, it will be proved that these points need to be updated to the new polygon model and wireframe model. Subsequently, the vertex is reorganized into a data structure of the tangent point and is added to the Map of the outer surface. Then, the 12 edges of the hexahedron are traversed to obtain the beginning and the end of the edge smooth function F(x) values; if the multiplication of two values is less than zero, this will imply that cutting is occurring on the edge, and the edge is obtained on the two endpoint coordinates $pos_0$ and $pos_1$. The function F(x) values $f_0$ and $f_1$ are smoothed; therefore, we can figure out the tangent point location on the edge of ratio coefficient of $u$ by the following formula:

$$u = \frac{0 - f_0}{f_1 - f_0}. \tag{17}$$

Then, the coordinate *pos* of the tangent point is generated by interpolation:

$$pos = pos_0 * (1 - u) + pos_1 * u. \tag{18}$$

Finally, the tangent point is added to the cutting surface Map and the corresponding outer surface Map.

After performing single mesh cutting, it will be known which vertices of the outer surface need to be redrawn in the process of cutting, and discrete tangent points are generated. Then, the discrete tangent points are triangulated.

## 8. Discrete Tangent Point Triangulation

After obtaining the discrete tangent points of each face in the same grid, the new generated tangent points of each face must be triangulated in the same grid with the original vertices that need to be redrawn. The triangulation must be completed immediately after each mesh has been cut; otherwise, the profile will be inconsistent, which will also have an impact on the contour display. The ultimate objective of discrete tangent triangulation is to obtain a sequence that represents the order of the edges of each point. According to this sequence, the corresponding points can be observed and then triangulated. There are only two cases that need to be triangulated for the points generated after cutting. The first case is the section, and the second case is the outer surface. After cutting, the points on the new surface will include both the vertices of the original mesh that need to be redrawn and the newly generated tangent point, and all points on the section are newly generated tangent points. Section triangulation and external surface triangulation will be analyzed below.

### 8.1. Section Triangulation

To generate the section, these discrete points must be triangulated. Surface cutting of the solid model can be regarded as cutting every small mesh plane and then constructing the surface from all microplanes. The number of tangent points generated by a plane to a hexahedron can only be three, four, five or six. That is, only triangles, quadrangles, pentagons and hexagons can be generated after cutting. As shown in Figure 5, it is easy to understand the case where the triangle and the quadrilateral are obtained after cutting, and the condition for obtaining the pentagon after cutting is that the tangent point must satisfy the following conditions: (1) a point on an edge; (2) points of two edges that are parallel to the edge; (3) the midpoints of two edges that on the vertical plane of the very edge and that do not intersect the very edge. The plane formed by the five points is used to cut the hexahedron to obtain a pentagon. The condition for obtaining a hexagon after cutting is that the tangent point must satisfy the following conditions: (1) first take a vertex; this vertex leads to three edges, taking the midpoints of the two edges that are not perpendicular to the bottom and top surfaces; (2) take the midpoint of the two edges that intersect the two edges of (1); (3) middle points of two edges which are parallel ones on the parallel faces of the two edges of (1) and any two of parallel ones are not intersected by another edge. On traversing these six points, a hexagon is obtained.

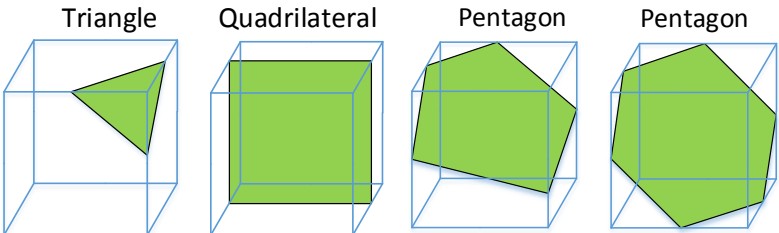

**Figure 5.** Possible shape after plane cutting hexahedron.

The process of discrete tangential triangulation after cutting is briefly described using examples of the above four cases. Four tangent points are taken as an example for the execution after the

cutting algorithm has completed. The tangent points are generated on four edges of hexahedrons that are numbered 0, 1, 8, and 9. These four tangent points form two triangles. The correct sequence of triangulation is 0, 1, 8, and 9. That is to say, the first triangle is composed of the tangent points on edges 0, 1, and 8, and the second triangle is composed of tangent points on edges 1, 8, and 9. The same applies to the pentagonal and hexagonal shapes after cutting. In this study, the order of cut points is specified by using a hash graph method to make the triangulated section smoother and avoid the intersection of section void and graph.

To complete triangulation, the order of connection of discrete tangents must be defined. Execution is simple for three points, but it is not so for four, five, or six points. To speed up the algorithm, the hash graph method is adopted, and the concept of space for time is utilized to optimize the algorithm speed in less space. Three Maps are defined for three different cases where the number of cut points is 4, 5 and 6 after cutting. Among them, there are 15 cases of four tangential points, 24 cases of five tangential points, and four cases of six tangential points. Only the correct order of these 43 cases needs to be saved in advance to meet all requirements of triangulation. KEY in the Map mentioned above can be expressed as:

$$KEY = \sum_{i=0}^{n} 2^{VALUE[i]}. \tag{19}$$

It represents the value converted from binary to decimal by each edge. If the tangent point is generated on an edge, the corresponding position of the edge is 1; otherwise, it is 0. In the above formula, *VALUE* is an array composed of edge numbers. Those arrays indicate the order of topological connection between edges, and $n$ is the number of *VALUE* data. For example, assuming that the number of tangent points after cutting is 5, and the tangent points are located on the five edges numbered 0, 2, 7, 8, and 11, the *KEY* value of this type of circumstance is shown in Figure 6. Because the hexahedron has 12 edges, it corresponds to 12 bits, the bottom right square represents the edge numbered 0, the left square represents the edge numbered 11, and the location of the occurrence of corresponding cutting edges is 1. Subsequently, a decimal *KEY* value of 2437 is obtained, which is in accordance with the topology constituting the three triangles, i.e., 0-2-8, 2-8-7, 2-8-7.

| KEY | VALUE | | | | | The bitwise representation of the cutting edge | | | | | | | | | | | |
|---|---|---|---|---|---|---|---|---|---|---|---|---|---|---|---|---|---|
| 2437 | 0 | 2 | 8 | 7 | 11 | 1 | 0 | 0 | 1 | 1 | 0 | 0 | 0 | 0 | 1 | 0 | 1 |

**Figure 6.** Schematic of the cutting situation.

To generate the section, we first need to initialize points four, five, and six of the tangency condition $Map_i$ and then calculate the *KEY* value according to the sequence of the edge of the tangent point and locate the correct triangulation sequence through the *KEY* according to the sequence. Subsequently, we identify the corresponding tangent point according to the order in the tangent point array to complete a single grid of triangulation. Smooth sections are generated after all meshes are triangulated.

*8.2. External Surface Triangulation*

To increase the diversity of visualization results and facilitate multi-angle observation of the cut data, we have also achieved the drawing of the cut entity; thus, we need to triangulate the points on the cut surface. The cut surface is special, and it composes vertices and tangent points that must be redrawn. If new data structures and algorithms need to be reorganized for each triangulation, the efficiency will decrease, and code can not be reused, resulting in redundant code. Therefore,

the vertices of the outer surface are processed, and the process of triangulation of the outer surface is directly transformed into the process of triangulation of the profile.

As mentioned above, to complete the triangulation of the section, we require the correct topological order of the edges where the tangent points are located; most importantly, we must determine the number of the edges where the tangent points are located. However, the outer surface is composed of newly generated tangent points and original vertices. The tangent points can be easily determined by the number of the edge on which the original vertex is located, and the original vertex is not on any edge. Therefore, the ultimate objective of the algorithm is to transform the original vertex into the data structure of the tangent points and determine the number of the most suitable edge for which the point is located. Subsequently, the method of section triangulation described in the previous section can be applied to achieve outer surface triangulation.

When the number of points in the outer surface array is exactly three, triangulation is performed directly on the three points; however, other cases require further analysis. The newly generated tangent points are not vertices in the grid and are marked $-1$, depending on the number of the vertices above. Similarly, the original vertex in the grid does not belong to any edge alone, so the edge number of the original vertex in the grid is marked as $-1$. After organizing and initializing the original vertices and tangent points of the mesh, the outer surface triangulation algorithm is implemented further.

Obtain the array of points in the Map of the outer surface. Identify which points belong to the original point and which points belong to the tangent point, and in the attribute of tangent point, obtain the number of points in the grid where the points that are located complete the above distinction. If the number is $-1$, it means that it is the tangent point; otherwise, it is the original point. When all points are original vertices, the triangulation sequence can be determined according to the number of the current face and the vertex number. When there are both original points and newly generated tangent points, we first obtain the number of edges that each original point may belong to. One point corresponds to three possible edges, and the corresponding edges of each original point constitute *possLineSet[i]* of possible edges. We define a set of edge numbers to be deleted *toBeDeleteSet*, and then identify the possible edges obtained before. The intersection between two edges can be obtained as follows:

$$intersectionSet = \bigcap_{i=0}^{nums} possLineSet[i]. \tag{20}$$

Then, we obtain the IDs of all tangent point edges and the set of all tangent point edges:

$$cutLineIDSet = \bigcup_{i=0}^{cnums} cutLineID_i. \tag{21}$$

In Formula (21), *cnums* represents the number of tangent points, *cutLineID$_i$* represents the ID of the corresponding edge of each tangent point, and the *intersectionSet* is merged with the set of all tangent points *cutLineIDSet* to obtain the set of edge numbers to be deleted:

$$toBeDeleteSet = intersectionSet \bigcup cutLineIDSet, \tag{22}$$

$$LineID = possLineSet[i] - toBeDeleteSet. \tag{23}$$

*LineID* in Formula (23) indicates that the original vertex has been constructed as a tangent point. After obtaining the number of the edge of this point, surface triangulation can be completed by directly performing the section triangulation algorithm.

## 9. Experimental Results and Analysis

The computer used in the experiment includes an Intel Core i7 3.6GHz CPU, NVIDA GeForce GT 630 GPU, with 4 GB memory and 2 GB display memory. In our experiment, we use OSG as rendering engine, and C++ and OpenGL as a programing language.

### 9.1. Cutting Effect

In the three-dimensional visualization technology, user interaction experience has become a very important indicator. Traditional cutting mostly realizes plane cutting. In this study, the method of drawing an arbitrary curve on a screen is proposed to complete surface cutting. This interactive mode brings better user experience. In order to verify the validity of this method, we have done plane cutting and surface cutting for dam II and the lead block models. In order to better show the effect model, we use solid and wireframe to render in the online frame mode. The visualization effect after cutting is real and comprehensive. It can not only make the shape of the cut entity clear at a glance, but also clearly see the topological relationship between all the tangent points of the section. Figures 7 and 8 represent the result after the plane cutting of dam II and the lead block, respectively, while Figures 9 and 10 represent the result after curve cutting of dam II and the lead block, respectively. In these figures, the original model before cutting is on the left, the solid effect picture after cutting is in the middle, and the section after cutting is on the right. As seen from the figures below, the plane equation used in plane cutting is displayed in the shown form of a red plane, which can draw the entity and the plane after cutting. The red free curve in the surface cutting represents the trajectory of the mouse, which can clearly draw the solid and cutting surface after cutting. As shown in Figures 9 and 10, the surface is very smooth after cutting. The values after cutting are calculated via interpolation and displayed in different colors; the points with large values are in red, while those with small values are in blue. In addition, the external surface of the cut entity is marked with black lines to identify the topological relationship of the finite element data, and the section profile is clearly divided in the form of a black outer frame, which is convenient for the user to conduct relevant operations on the cut entity and complete their experimental analysis with more intuitive profiles.

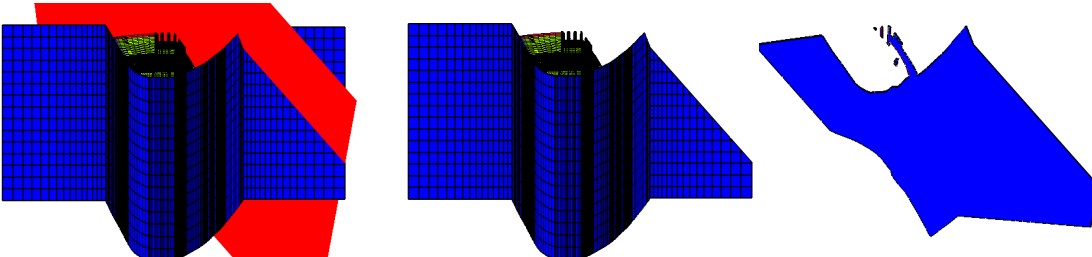

**Figure 7.** Traditional plane cutting result of dam II.

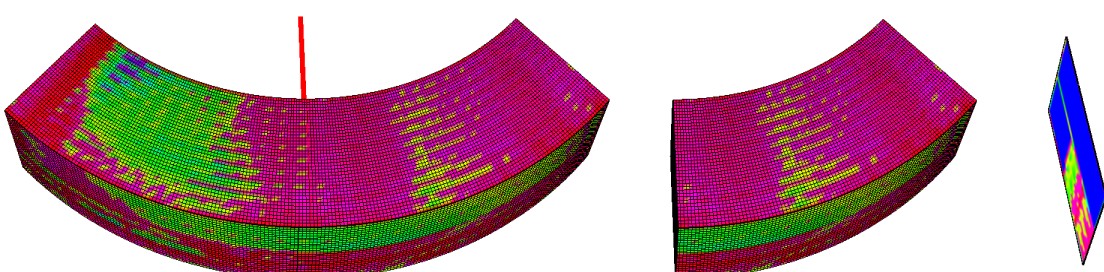

**Figure 8.** Traditional plane cutting result of the lead.

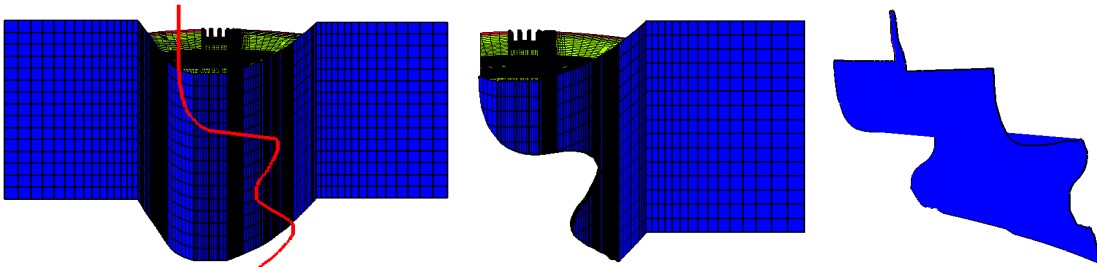

**Figure 9.** Our free curve cutting result of the dam II.

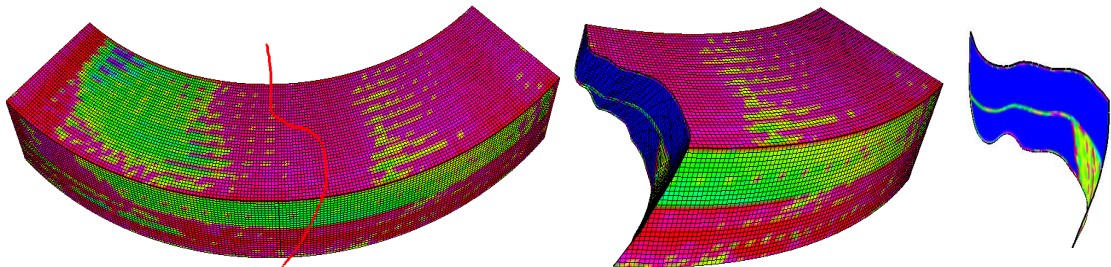

**Figure 10.** Our free curve cutting result of the lead.

*9.2. Cutting Response Time*

Several experiments were conducted on different models, and the interaction time between plane cutting and surface cutting was finally obtained, as shown in Table 1. Clearly, for millions of data points, the cutting display can be completed in approximately 10 s; for 100,000-level data, the response time before and after cutting can be reduced to within 5 s. For 10,000-level data, the response time before and after cutting is less than 1 s. The response time can meet the needs of the vast majority of users interaction, greatly improve the user experience, and considerably improve the work efficiency of engineers. The cutting time of the surface is often greater than that of the plane because plane cutting is in the form of the directly specified equation, whereas surface cutting requires the use of the interpolation method based on RBF to calculate the smooth equation, and the process of constructing the smooth equation consumes part of the time.

**Table 1.** Cutting time for different objects.

| Entity Name | Vertex Number | Mesh Quantity | Plane Cutting Time | Curve Cutting Time |
|---|---|---|---|---|
| shock absorber | 2301224 | 2155840 | 13514 ms | 21941 ms |
| dam I | 659981 | 621902 | 3780 ms | 6152 ms |
| dam II | 323859 | 293794 | 1805 ms | 3133 ms |
| lead | 211437 | 197600 | 1207 ms | 2034 ms |
| experiment table | 87237 | 79928 | 596 ms | 927 ms |
| shear wall structure | 52865 | 24370 | 354 ms | 619 ms |
| joist | 40164 | 25256 | 296 ms | 429 ms |

## 10. Conclusions

To display the internal data of finite element data easily, a method for generating the arbitrary sections of finite element data based on radial basis function (RBF) interpolation is proposed in this paper. We provide a simple, effective method for displaying the internal information volume data. The experimental results have validated that the section generated by our method is correct. In addition, the efficiency of our method is good enough for the users. The method not only completes the cutting by a drawing free curve but also improves the interaction experience greatly.

From the field of application, our method can be widely applied in engineering analysis, medical diagnosis, computer animation and so on. To further improve its efficiency, we will try to accelerate our method based on GPU in the future. This will further improve the user interaction experience.

**Author Contributions:** Methodology, Y.Z.; Project administration, S.X.; Software, F.X.; Validation, X.L.; Writing—review and editing, L.C.

**Funding:** This research was funded by the National Key R&D Program of China (No. 2018YFC0407102), the China Institute of Water Resources and Hydropower Research (IWHR) Basic Research Fund (No. SM0145B632017, SM0145B952017, SM0145C102018), and the Open Research Fund of State Key Laboratory of Simulation and Regulation of Water Cycle in River Basin, IWHR (Grant No. SM0145B252018).

**Acknowledgments:** The authors appreciate the assistance of the anonymous reviewers for their constructive suggestions for improving the quality of this paper.

**Conflicts of Interest:** The authors declare no conflict of interest.

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
