# Peer review of "Visualization Method for Arbitrary Cutting of Finite Element Data Based on Radial-Basis Functions"

_information, doi:10.3390/info10070229_

Reviewer 1 Report

The paper aims to present a method for generating the arbitrary section of finite element data based on the RBF interpolation functions. The topic is interesting and well-developed in the manuscript. Generally speaking, the paper is clear and could be accepted for publication. A minor revision is suggested to improve the following aspects:

1. Some typos should be checked and corrected throughout the whole manuscript. The use of English could be improved as well.

2. Section 4 is extremely concise, and some aspects are not so clear. Please, revise this section by adding further comments and details about the interaction mode.

3. The theoretical background of RBF could be introduced in a more complete manner. Please, revise this aspect. The authors could mention also some additional examples of RBF applications. The list of references could be improved by mentioning further contributions related to RBF.

4. The “Conclusion” section should be enhanced and enlarged. The authors could emphasize more the main achievements of the research and could also mention some practical (numerical) applications in which this technique could be required.

Author Response

Thank you for your comments on my article. I have made some modifications to the article in response to these problems. Details can be found in the annex.

Reviewer 2 Report

Summary of the paper: The paper deals with FEM-visualization, more specifically with how to efficiently visualize arbitrary cross-sections through a body (as opposed to planar cross sections). The authors use radial basis functions (RBF) to interpolate pixels drawn by a user into a smooth function. This function is then used to determine how to cut through the finite element mesh. The finite elements are altered by introducing planar cuts through them, according to the RBF-function. The method is demonstrated by some experiments measuring the difference in visualization time between making a planar cut and an arbitrary cut. Speaking from the point of view of a visualization software user, the topic of efficient arbitrary visualization of cross sections is highly relevant.

Major points to improve:

Unfortunately the paper is very difficult to understand due to the language. I strongly suggest sending the paper to a professional language review. I will therefore not comment on language in this review.

I am missing a clear overview of other interpolation methods used in the field, what their caveats are and why RBF is better. If there is no other interpolation methods for arbitrary cuttings, that should be made clearer.

In the experiments, I would like to see a comparison to other interpolation functions used, or other softwares, to explain why this method is better. It is not so informative to compare planar cross-sections with the arbitrary ones.

In section 6, 7, and  8, it is not clear to me if this the standard approach or new methods

Minor points to improve:

The paper has very nice figures that I appreciate, but I would prefer some more descriptive figure captions.

In the experiment section, I am missing a short explanation of each of the experiment set ups.

In the conclusion, you mention that there will be no cavities. I would like some discussion on cavities earlier in the manuscript to understand in which cases cavity appears.

For getting the attention of visualization software users, it would be interesting to have some examples of different methods used in different softwares, such as ParaView

Author Response

(The authors gave the same response as above.)

Reviewer 3 Report

This paper introduces the Radial Basis Function (RBF) to construct a smooth interpolation curve from user-input discrete points. The curve then is used to cut the finite element model for interior data visualization. Comments and suggestions are given as follows.

The novelty of the current manuscript as the authors highlighted is the use of RBF for construction of a smooth curve from user-input points. However, the RBF is a global function, leading to a dense system of equations for obtaining required coefficients. Many local interpolation functions that are well suited for the purpose mentioned have been developed and widely used in the CAD/FEM field. The reviewer does not see convincing evidence that the proposed RBF cutting scheme is more effective than the existing schemes based on local interpolation function, such as B-spline or even the Finite Element shape functions that are used in the analysis. In general, more scientific evidence needs to be provided to demonstrate the effectiveness of the method, including, numerical stability (conditioning of the system for solving coefficients) and sensitivity to the input data. A comparison between the proposed method and existing ones should be included.

Once the RBF curve is constructed, the authors proposed the algorithms/procedures for surface cutting of the FEM model, triangularization of the section, and transforming FEM data onto the section. These procedures and associated algorithms have been well developed and applied in the FEM\CAD industry. The authors should highlight the new development or contributions and/or what are the unique features when the RBF curve is used.

It is suggested to improve the overall presentation of the paper. The equations provided in the paper are mostly trivial, providing minimum information, whereas the key algorithms, e.g. cutting, triangularization, projection, and data transformation, are described vaguely by lengthy wording. Along with frequent grammatical and punctuation errors, it is difficult to grasp the key ideas and contributions of the paper.

The index in equations (6) and (7) is confusing. According to Figure 3, the reviewer wonders if the authors intended to express

c2i-1 = pi + αqi

c2i = pi + α q^i

 In Table 1, what are those models, “shock absorber”, “dam I”, … etc.? The authors may consider putting these models in Appendix or as the supplementary data.

Author Response

Thank you for your comments on my article. I have made some modifications to the article in response to these problems. Details can be found in the annex.

Round  2

Reviewer 1 Report

The current version of the manuscript is suitable for publication. Adequate answers and modifications have been included in the revised paper. Therefore, it can be accepted for publication in the present form.

Author Response

Please see the annex.

Reviewer 2 Report

The manuscript language is improved and I appreciate the authors answers to my concerns. However, I would still like to see an overview of methods others use for arbitrary cutting. Specifically, ellaborate on this section:

"Although there are some methods to obtain arbitrary cutting surfaces, most of them need professional operations. So the interaction is very tedious, and the cutting surface cannot be obtained quickly" and relate your method to the methods used in Zhang and Takayama et 91 al.[18,19]

As the authors cannot run experiments to compare with other methods I think it is important to clearly compare your approach to others at least in the text.

Author Response

Please see the annex.

Reviewer 3 Report

The authors properly addressed the reviewers' concerns. It is recommended to be published.

Author Response

Please see the annex.
